# Simulation and Experimental Study of the Influence of the Baffles on Solar Chimney Power Plant System

**Haixia Wang** [1]**, Jusheng Chen** [1]**, Ping Dai** [1]**, Fujiang Zhang** [2] **and Qingling Li** [1,*]

1   College of Electromechanical Engineering, Qingdao University of Science and Technology,
    Qingdao 266061, China; wanghaixia_1981@163.com (H.W.); jachin723@163.com (J.C.);
    daiping936@163.com (P.D.)
2   Zhongyuan Transportation Oil & Gas Company, National Pipeline Network Group North Pipeline Co. Ltd.,
    Dezhou 253000, China; zfjwhx@126.com
*   Correspondence: edu@qust.edu.cn

**Abstract:** The solar chimney power plant system (abbreviated as SCPPS) is a clean and pollution-free facility for generating electric power. To improve the generating efficiency, a bank of baffles can be arranged under the collector in SCPPS. ANSYS Fluent 18.2 was used to numerically simulate 3D models of SCPPS with or without baffles, and an experimental apparatus was built for verification. There are seven different types of model discussed here: the SCPPS without baffles (prototype), and other six types of models with different baffles (a-type, b-type, c-type, d-type, e-type, and f-type). The pressure fields, temperature fields, velocity fields, and power outputs of different models under the different baffles are discussed. It is shown that the addition of baffles in the system can increase the temperature field, pressure field, velocity field, and power output to varying degrees, but b-type baffles better improve the temperature and velocity uniformity of the system, and intensify the output power. For b-type, the simulation of the systems with five different baffle numbers (3, 4, 6, 8, and 12, respectively) was carried out, and it was concluded that the system with 12 baffles is the best in improving the system performance. It can be seen that the more the number of baffles, the better the performance of SCPPS. The experiments are also verified the simulation.

**Keywords:** SCPPS; baffles; velocity uniformity; output power

## 1. Introduction

SCPPS is a simple, clean, and pollution-free facility, by which the solar energy can be converted to electricity. Solar energy passes through the collector, then the heat storage layers are heated, thus the air under the collector are heated up. The density of the heated air decreases and flows upward along the chimney to force the rotation of the shaft of the turbine at the bottom of the chimney, and the turbine drives the generating set to generate electrical power. With the low power generation efficiency and large occupied area of the system, it is difficult for large-scale development. For these problems, many scholars have researched optimization studies on its structure.

Solar chimney power technology did not draw researchers' attention until Professor Schlaich proposed this technology in a conference in 1974, and in 1978 the world's first solar chimney power test plant began to be established in Spain and ran successfully for seven years after the completion, with an output power of 50 kW [1–3]. In 2002, Jiakuan Yang et al. [4–7] established a small SUPPS experimental device in Wuhan, China, and the output power was 0.003 W. In 2018, Ajay Bejalwar [8] built an experimental device covering 250 square feet area, and measured 10–25 W generated power without other energy sources. In 2020, Dara Khalid Khidhir et al. [9] increased the temperatures at the chimney entrance by adding a tracking reflector, and concluded that the output power of the new test system increased by 56.867% compared with the original system. However, the cost of experimental research is relatively high. With the continuous development of

simulation technology, it has gradually become a trend to use simulations or simulations combined with experiments to optimize the system.

Jing Nie et al. [10] analyzed the heat efficiency of the system under different inclination angles by introducing the collector influence coefficient d, and indicated that the SCPPS with the inclination angle 10° collector is the most economical in the Hohhot area. Qilong Cai et al. [11] studied the inclination angle of the collector under a fixed area, and it was shown that the performance of the system increases first and then decreases with the increase of the inclination angle. Hailin Zhu et al. [12] analyzed and calculated the collector radius and the relationship among the air flowing velocity, the convection heat transfer coefficient, and the heat absorption efficiency in the system. Koonsrisuk et al. [13] indicated that the power generation will be increased when the entrance area of the collector is smaller than the exit area.

Pritam Das et al. [14] compared and analyzed the influences of different chimney angles on the gas flow characteristics of the solar chimney by numerical simulation. They found that when the chimney angle is 2°, the gas velocity, pressure gradient and air volume flow rate of this system are the largest, and the system output power and total efficiency are the highest. Erdem Cuce et al. [15] simulated and studied the effect of chimney height on SCPPS performance using the model of the Manzanares test plant. Huilan Huang [16] analyzed and studied the effects of different heights, diameters and inclination angles of the chimney on the efficiency of the solar chimney, and concluded that with the height and diameter of the chimney increasing the efficiency of the system increases. When the inclination angle increases to a certain value, the efficiency of the chimney does not change. Yangyang Xu et al. [17] designed and studied an expanded-type SCPPS, which intensified the power output by increasing the area ratio of the chimney outlet to inlet in a larger range.

Pastohr et al. [18] considered the turbine as a reverse fan, and assumed the pressure drop at the inlet and outlet of the turbine for simulation research based on the Bates theory. Yisheng Chen et al. [19] calculated the turbines with different numbers of blades in SCPPS, and comprehensively compared the advantages and disadvantages of different numbers of blades. TP Fluri et al. [20] studied different configurations of turbines and concluded that the efficiency of the system with single vertical axis turbine is better than other configurations. Tingzhen Ming et al. [21,22] designed and studied the 5-blade turbine, and found that the increase of turbine velocity would increase the air outlet temperature and pressure drop, whereas the outlet average air velocity and flow rate would reduce.

Li Zhu et al. [23] designed an experimental apparatus of solar chimney for different heat storage methods and measured the temperature distribution and temperature variation performance in the solar chimney. Lu Zuo et al. [24] built the experimental apparatus with heat storage layers with different stones, compared and analyzed the influence of heat storage layer of different stones on the performance of the solar chimney. Huilan Huang et al. [25] used three different materials, sand, stone, stone, and water-filled black sealing pipes, as a heat storage layer. Through experiments, it was concluded that the third has the best heat storage performance, increases the temperature difference between the inlet and outlet of the system, and raises the heat collection efficiency of the solar chimney. Sciuto, G.L., et al. [26] conducted a simulation study on the hybrid energy storage system on account of the cross comparison of finite element and neural network, and proposed a method to predict the system performance.

In 2002 Anthonony [27] proposed installing inlet guide vanes in front of the turbine. These inlet guide vanes can make the air pre-swirl flow before it enters the turbine. Tingzhen Ming [28] simulated the Manzanares SCPPS prototype and found that adding a plate on the soil could have a significant impact on the local relative static pressure and velocity, increasing the output power of SCPPS. Siyang Hu [29] carried out a simulation study on the diversion cone at the bottom of the chimney with the same shape but with different aspect ratios. It was shown that the diversion cone with the best length-diameter ratio may significantly improve the flow field of the system. Huizi Li et al. [30] simulated a SCPPS with eight spiral guide plates, and found that the model can effectively reduce the

floor area with the same output power. Tingzhen Ming et al. [31] set eight radial baffles in the sloped SCPPS. It is found that adding baffles can effectively use wind energy and reduce hot air spills, therefore increasing the output power of the system.

In summary, adding a bank of baffles (or pre-swirling plates) under the collector can guide the air flow, and will affect the air flow and heat transfer, and may change the flow and heat transfer performance of the system. The scholars in the world have mentioned baffles, but there are a few detailed studies on the optimization of the types and numbers of baffles. To further improve the power generation efficiency of SCPPS under the same radial area, in this paper different types and numbers of baffles in the system are discussed and analyzed. The output power and efficiency of the SCPPS with or without baffles are simulated and compared, the improvement effect of the baffles. Compared with different influences of six different types of baffles on the system, the best baffle type is ascertained. Compared with different SCPPS with different numbers of baffles, the optimum number of baffles is selected. Finally, the experiments on SCPPS with or without baffles are verified the simulation conclusion.

## 2. Numerical Simulation

### 2.1. Physical Model

The 3D model is based on a 1:1 scale of the Manzanares SCPPS test model [32,33]. The radius of the collector is 122 m, the entrance height is 2 m, the height of the chimney bottom center above the ground is 6 m, the overall chimney height is 194.6 m, and the chimney diameter is 10.16 m. The size of the model is shown in Figure 1a, and 3D model can be seen in Figure 1b.

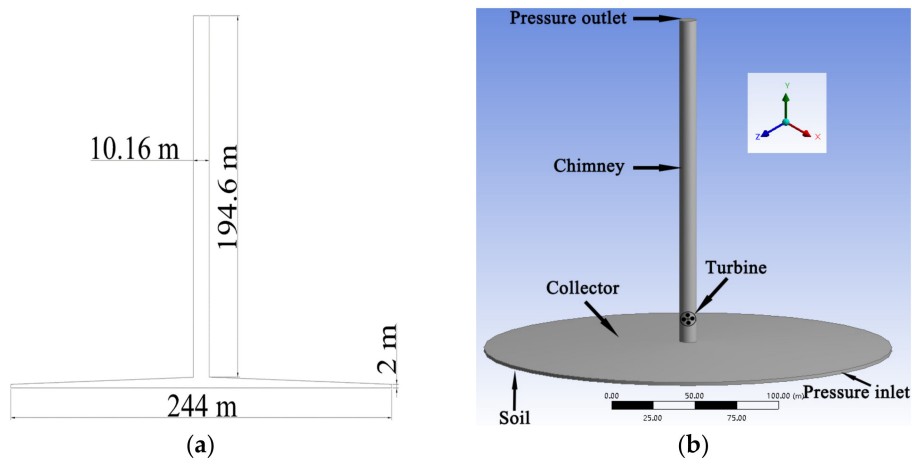

**Figure 1.** SCPPS model. (**a**) Sizes of the model; (**b**) 3D model.

The baffles of different shapes are distributed along the vertical ground direction between the collector and the soil. The dimensions of the baffles are shown in Table 1. The baffles of different types are shown in Figure 2, and the thicknesses of the baffles are ignored.

**Table 1.** The dimensions and types of the baffles.

| Parameters | a-Type | b-Type | c-Type | d-Type | e-Type | f-Type |
|---|---|---|---|---|---|---|
| Scale | Full-scale | Half-scale | Half-scale | Full-scale | Half-scale | Half-scale |
| Location | - | inner | outer | - | inner | outer |
| Length | 116.92 m | 58.46 m | 58.46 m | 116.92 m | 58.46 m | 58.46 m |
| Radius of curvature | - | - | - | 130.72 m | 130.72 m | 130.72 m |
| The height of the baffle near the entrance inlet | 2 m | 2.88 m | 2 m | 2 m | 2.88 m | 2 m |
| The height of the baffle near the chimney inlet | 5.75 m | 5.75 m | 2.88 m | 5.75 m | 5.75 m | 2.88 m |

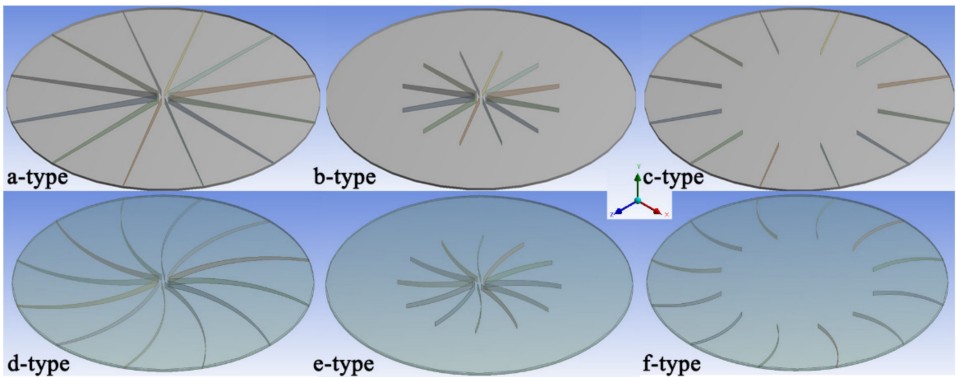

**Figure 2.** Schematic diagram of different types of baffles.

*2.2. Mathematical Model*

To simplify the calculation, some assumptions about the model are made:

(1) The air flow in the model is at steady flow.
(2) There is no heat source.
(3) The heat transfers between the flowing air and the system is considered without regard to the heat transfer between the environment and the system.
(4) The material properties of the material are constant.

The air flow in SCPPS is usually regarded as natural convection, and its flow state is judged by the Rayleigh number *Ra* [34].

The Rayleigh number *Ra*:

$$Ra = \frac{\beta g \rho \Delta T L^3}{\nu \alpha} \tag{1}$$

Here, $L$ is the characteristic length; g is the acceleration of gravity; $\rho$ is the fluid density; $\nu$ is the fluid kinematic viscosity; $\alpha$ is the thermal diffusion coefficient.

If the Rayleigh number is greater than $10^{10}$, the flow belongs to the turbulence [35]. From calculations it was concluded that the air flow of SCPPS is turbulent.

For the natural convection flow, Boussinesq Assumes of the fluid can effectively be dealt with the relationship between fluid temperature and its density changes. In this model the air density can be treated as a constant value in all solved equations, except for the buoyancy term in the momentum equation:

$$\rho = \rho_0 (1 - \beta \Delta T) \tag{2}$$

where $\rho_0$—the density of the fluid at standard conditions; $\beta$—thermal expansion coefficient of fluid; $\Delta T$—Temperature difference in the system.

In particular, this approximation is accurate as long as the changes in actual density of fluid are small. When $\beta \Delta T << 1$, Boussinesq approximation is valid.

The flow of a fluid must follow three laws of conservation of mass, conservation of momentum and conservation of energy [36,37], and the actual flow process of a fluid can be represented by the three basic equations.

Continuity equation:

$$\frac{\partial \rho}{\partial t} + \frac{\partial(\rho u)}{\partial x} + \frac{\partial(\rho v)}{\partial y} + \frac{\partial(\rho w)}{\partial z} = 0 \tag{3}$$

The flow of the fluid is assumed to be steady. Combined with Equation (2), it can be known that at any time the density is constant. Therefore, $\partial t = 0$, and Equation (3) becomes:

$$\frac{\partial(\rho u)}{\partial x} + \frac{\partial(\rho v)}{\partial y} + \frac{\partial(\rho w)}{\partial z} = 0 \tag{4}$$

Momentum equations:

$$\frac{\delta(\rho u)}{\delta t} + \frac{\delta(\rho uu)}{\delta x} + \frac{\delta(\rho uv)}{\delta y} + \frac{\delta(\rho uw)}{\delta z} = -\frac{\delta p}{\delta x} + \mu(\frac{\partial^2 u}{\partial x^2} + \frac{\partial^2 u}{\partial y^2} + \frac{\partial^2 u}{\partial z^2}) \tag{5}$$

$$\frac{\delta(\rho v)}{\delta t} + \frac{\delta(\rho vu)}{\delta x} + \frac{\delta(\rho vv)}{\delta y} + \frac{\delta(\rho vw)}{\delta z} = -\frac{\delta p}{\delta y} + \mu(\frac{\partial^2 v}{\partial x^2} + \frac{\partial^2 v}{\partial y^2} + \frac{\partial^2 v}{\partial z^2}) \tag{6}$$

$$\frac{\delta(\rho w)}{\delta t} + \frac{\delta(\rho wu)}{\delta x} + \frac{\delta(\rho wv)}{\delta y} + \frac{\delta(\rho ww)}{\delta z} = -\frac{\delta p}{\delta z} + \mu(\frac{\partial^2 w}{\partial x^2} + \frac{\partial^2 w}{\partial y^2} + \frac{\partial^2 w}{\partial z^2}) + \rho g \beta \Delta T \tag{7}$$

Energy equation:

$$\frac{\delta(\rho cT)}{\delta t} + \frac{\delta(\rho cuT)}{\delta x} + \frac{\delta(\rho cvT)}{\delta y} + \frac{\delta(\rho cwT)}{\delta z} = \lambda(\frac{\partial^2 T}{\partial x^2} + \frac{\partial^2 T}{\partial y^2} + \frac{\partial^2 T}{\partial z^2}) \tag{8}$$

where $u$, $v$, $w$ is the fluid velocity component in $x$, $y$ and $z$ directions respectively; $t$ is time; $c$ is the fluid specific heat at constant pressure; $T$ is the fluid temperature; $p$ is the fluid pressure; $\mu$ is the fluid dynamic viscosity; $\lambda$ is the fluid effective thermal conductivity.

Turbulent kinetic energy equation:

$$\frac{\delta(\rho k u_i)}{\delta x_i} = \frac{\delta}{\delta x_j}((\mu + \frac{\mu_t}{\sigma_k})\frac{\partial k}{\partial x_i}) + G_k + G_b - \varepsilon\rho + S_k \tag{9}$$

$$\frac{\delta(\rho \varepsilon u_i)}{\delta x_i} = \frac{\delta}{\delta x_j}((\mu + \frac{\mu_t}{\sigma_\varepsilon})\frac{\partial \varepsilon}{\partial x_i}) + C_{1\varepsilon}(G_k + C_{3\varepsilon}G_b) - C_{2\varepsilon}\rho\frac{\varepsilon^2}{k} + S_\varepsilon \tag{10}$$

In the equations above, $G_k$ is turbulent kinetic energy generated by average velocity gradient; $G_b$ is turbulent kinetic energy generated by buoyancy; $\varepsilon$ is dissipation; $C_{1\varepsilon}$ is 1.44; $C_{2\varepsilon}$ is 1.29; $C_{3\varepsilon}$ depends on the relationship between the direction of gas flow and the direction of gravity (If they are parallel, the value is 1.0; and if they are perpendicular, the value is 0); $\sigma_k$ is 1.0; $\sigma\varepsilon$ is 1.3.

Theoretical output power of the turbine:

$$P_{th} = (\Delta p \times Q) \tag{11}$$

where $\Delta p$—the pressure drop; $Q$—the volume flow rate.

Assuming that the conversion efficiency $\eta_{tur}$ of the turbine device is 66.7% [14], the actual output power of the turbine device:

$$P_{act} = P_{th} \times \eta_{tur} \tag{12}$$

### 2.3. Boundary Condition

To accurately simulate the flow characteristics of the model under sunlight conditions, the solar calculator in Fluent 18.2 is used to calculate the radiation value in China, Qingdao. Based on the location the solar azimuth vector is x = −0.230763, y = 0.952219, z = 0.200071. The standard k-ε equation is selected, and the wall surface is processed by the standard wall surface function. In that the air flow in the chimney is turbulent, and the standard the k-ε model which is used for the single-phase flow across the wall is developed to describe the air flow in SCPPS except the porous heat storage layers [38–41] as it was properly used to analyze the performance of solar chimney [38,42] and has been proved valid by the tests [43]. The coupling of pressure and velocity is calculated by SIMPLE (semi-implicit method for pressure linked equation) algorithm. To intensify the accuracy of calculation, double precision is used for simulation. In the pressure equation is adopted by Body Force Weighted. In the continuity equation, momentum equation, energy equation, turbulent kinetic energy and turbulent energy dissipation rate are all selected with Second order upwind.

The system wall boundary conditions are set as follows:

(1) The inlet is set as the pressure inlet and the ambient temperature is 20 °C;
(2) The outlet is set as the pressure outlet and the ambient temperature is 20 °C;
(3) The collector is used as the mix boundary, the thermal conductivity is 8, and the free flow temperature is set. The temperature is 20 °C the ambient temperature is zero degrees of insulation, and the reflectivity is 0.49. It is referred as translucent material in solar radial conditions;
(4) The baffles are set as the coupling wall conditions;
(5) The soil is assumed as the 40 °C heat storage layer;
(6) The other walls are all heat-insulated surface with no slippage.

### 2.4. Validation

2.4.1. Grid Independence Validation

In numerical calculations, the accuracy and reliability of the calculation results can be directly affected by meshing. The fluid flow in SCPPS is complicated. To accurately simulate the influence of the gas flow and the baffle structure on the system performance. The structured grids are used to mesh the model, and the grids are shown in Figure 3.

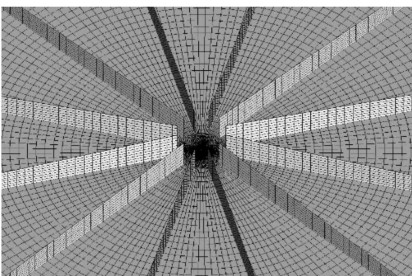

**Figure 3.** The structured grids of SCPPS with 12-piece a-type baffles.

To make certain the independence of the calculation results with the number of grids, different numbers of grids of the same physical model were calculated and simulated, and the velocities at the located surface of the turbine were selected as the variable to be verified for grid independence. The results are shown in Figure 4. According to the calculation results of different grid numbers, it is found that when the grid is densified to a certain extent, the velocity and the variable trend are similar on the whole. Considering the computational cost, the numbers of grids used by different models in the simulation are shown in Table 2.

**Table 2.** Number of grids used in different models.

| Type of Baffle | Number of Grids |
| --- | --- |
| prototype | 345,600 |
| a-type | 481,280 |
| b-type | 394,925 |
| c-type | 403,488 |
| d-type | 414,120 |
| e-type | 385,488 |
| f-type | 429,240 |

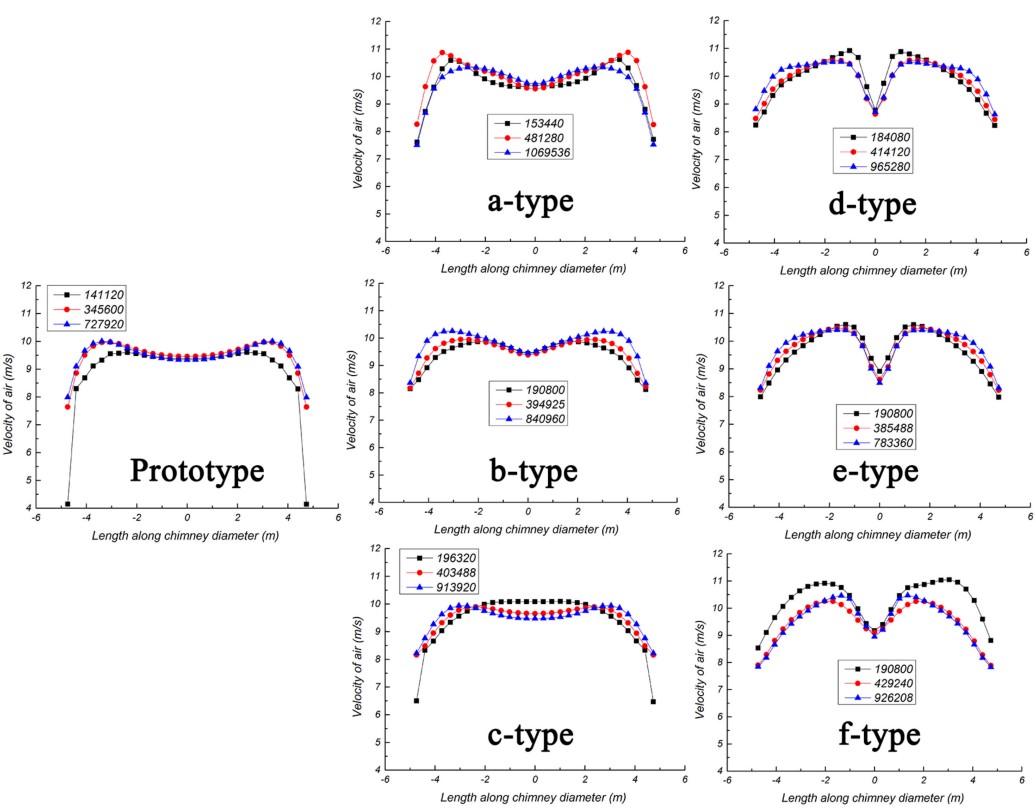

**Figure 4.** Grid independence verification.

### 2.4.2. Model Validation

To verify the model, the simulation results of the large model were compared with the Manzanares experimental model report [27], as seen in Table 3. The results show that this model can be used for the prediction of SCPPS performance. As shown in Table 3, the solar radiation intensity received is almost the same, and the difference in temperature, velocity and output power of the turbine blades region is 5.5 °C, 0.2 m/s and 1.5 kW, respectively. In this model the soil is considered to be a constant, while the soil gradually rises near 40 °C. The temperature of this model is higher than in the experiment, and the temperature difference between them is large, with the deviation rate of +30.1%. With the influence of the lower velocity, the deviation rate of the output power in this model calculated is +6.2% from the experimental output power. In consequence, this model can be used to predict SCPPS of similar sizes.

**Table 3.** Verification and comparison of the model results.

| Parameters | Manzanares Experiment [27] | This Simulation | ERROR Compared with the Experimental Data |
|---|---|---|---|
| Solar radiation intensity (W/m²) | 879 | 877 | 0% |
| Temperature (°C) | 18 | 23.5 | +30.1% |
| Air velocity around turbine (m/s) | 9.9 | 9.7 | −2.0% |
| Power (kW) | 24.2 | 25.7 | +6.2% |

## 3. Experimental Study

### 3.1. Experimental Apparatus

The miniature experiment apparatus is based on the Manzanares experimental plant on a scale of 1:100, shown in Figure 5, and its dimensions are shown in Table 4.

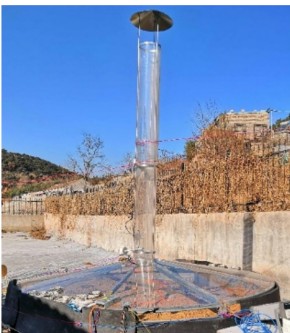

**Figure 5.** Miniature experimental plant of SCPPS.

**Table 4.** The dimensions of the miniature experimental plant.

| Sections | Dimensions |
|---|---|
| Height of the chimney | 2000 mm |
| Diameter of the chimney | 200 mm |
| Diameter of the collector | 2440 mm |
| Ground height of the collector | 600 mm |
| Inclination angle of collector | 8° |
| Thickness of the heat storage layer | 800 mm |

In view of the size of the experimental apparatus and conditions, the baffle is made of 4 mm acrylic. Three types of baffles, a, b, and c, are tested. The shapes and sizes are shown in Figure 6a. The distribution in the system is shown in Figure 6b.

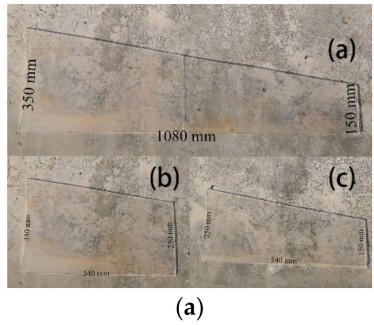

(**a**)

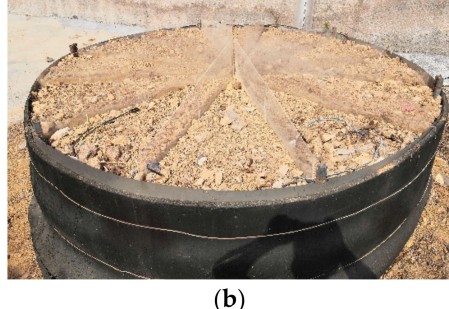

(**b**)

**Figure 6.** The shape and size and distribution of the experimental baffles. (**a**) a, b, c three types of baffles; (**b**) a-type baffle distribution.

The chimney is also made of perspex, and its solar transmissivity is about 85-92%. The collector is PMMA, its thickness is 5 mm. The overall frame of the experimental apparatus is welded with square tubes, and the area under the collector is soil and scree. The baffle is also chosen transparent acrylic considering its solar radiation penetration.

*3.2. Experimental Method*

In the experiment, the TES-1333 solar radiation meter was used to measure the solar radiation intensity. The data were recorded every minute in the experiment, and the average value was taken as the solar radiation intensity during the experiment. TES-1341 hot-wire anemometer was used to measure the entrance wind velocity of the collector and the Testo-435 multifunctional measuring instrument was used to measure the wind velocity and pressure at the center of the bottom of the chimney. The instruments measuring temperatures and velocities automatically recorded data at an interval of 10 s, with 99 data in a group and nine groups in one experiment. Temperature was measured by K-type thermocouples. There were 12 thermocouples distributed in the collector, and

two thermocouples are distributed at the bottom of the chimney. Temperature data were collected by Agilent data acquisition instrument. Temperatures were acquired at an interval of 10 s. The measuring points as shown in Figure 7.

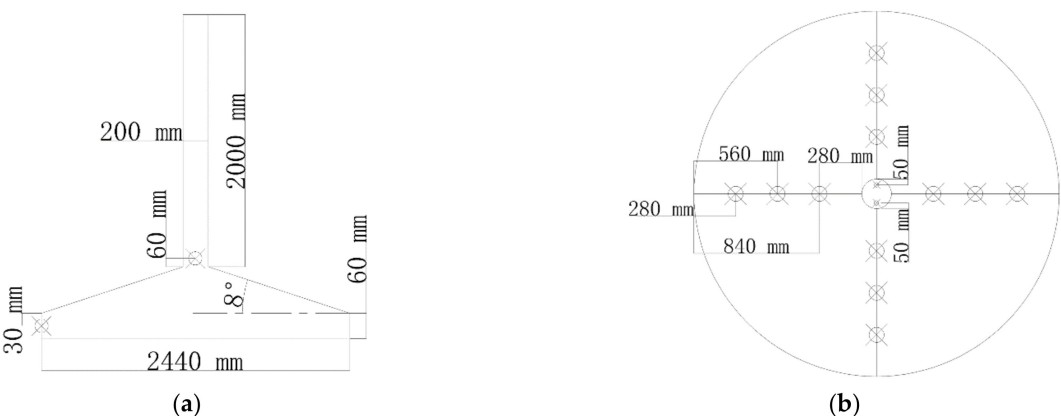

(**a**)    (**b**)

**Figure 7.** Schematic diagram of measuring point distribution of experimental apparatus. (**a**) Wind velocity measuring points; (**b**) Temperature measuring points.

The accuracy of the instruments used in the experiment varies. The measurement range of TES-1333 solar radiation meter is 0–2000 W/m$^2$, the resolution is 1 W/m$^2$, the accuracy is $\pm$10 W/m$^2$, and its temperature coefficient is $\pm$0.38 W/m$^2$/°C, TES-1341 Hot-Wire Anemometer has a range of 0.1–30.0 m/s, a resolution of 0.01 m/s, and an error of $\pm$3%. The test temperature range of K-type thermocouple is $-$40 °C + 375 °C, and the tolerance value is $\pm$1.5 °C. The 1-year DCV accuracy of the Agilent Date Acquisition is 0.004%, the resolution is 22 bits, and the measurement range is $-$100 °C to 1200 °C when a K-type thermocouple is connected, the accuracy is 1 °C, and its temperature coefficient is 0.03 °C. Due to the uncontrollable factors of the outdoor environment, the weather conditions where the maximum change in ambient temperature and solar radiation intensity does not exceed 20% is used in the experiment time.

*3.3. Experimental Validation*

The accuracy of the experimental data measured by the mini-type experiment apparatus of SCPPS need to be verified, thus used by the above-mentioned validated simulation method was established a numerical model as the same size of the experiment apparatus. The results of numerical computation and experiment are compared, shown in Figure 8.

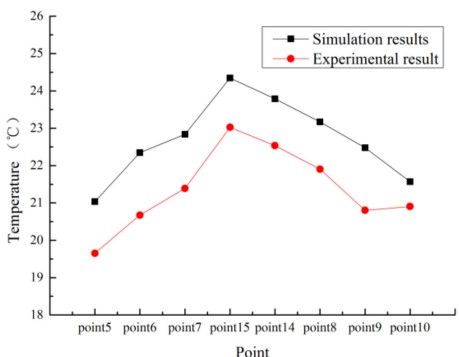

**Figure 8.** Comparison of the simulation and experimental results.

It can be seen from Figure 8 that the temperatures shown by the simulation are higher than the temperatures measured in the experiments. The stable boundary conditions are used in the simulation without regard to the influence of environmental factors, so the overall temperature distribution in the simulation is higher than the temperature measured

in the experiments. It is concluded that the temperature difference is 7.5% at most, 3.1% at least, and 5.9% on average. It is shown that both of the temperature distributions are the similar, which is gradually rising from the entrance of the collector to the entrance of the chimney, and the temperature at the bottom of the chimney is the highest. Thus, the experimental results can be considered to be reliable.

## 4. Numerical Simulation Results and Analysis

### 4.1. The Influence of Different Types of Baffles

#### 4.1.1. The Influence on the Temperature

In the SCPPS the air under the collector is mainly heated by absorbing solar radiation energy. The higher temperature of the air in the system, the more output power improved of the system. The ZOX plane at 1 m was taken under the collector, and compared with the temperature distribution of the system without baffles. The distributions of temperature in SCPPS are shown in Figure 9. It is shown that the temperature of the air rises gently from the entrance of the collector to the central chimney, and higher temperatures appear at the center of the chimney entrance. The entrance temperature in SCPPS with a-type baffles is the lowest, which is 294.6 K, while that with d-type baffles is the highest, which is 295.2 K, with a difference of only 0.6 K. The highest temperature of the plane in SCPPS with c-type baffles is the lowest, which is 300.0 K, while that of b-type baffles is the highest, which is 301.2 K, with a difference of only 1.2 K. The differences are tiny. Thus, the temperature distributions in SCPPS with different type baffles are similar.

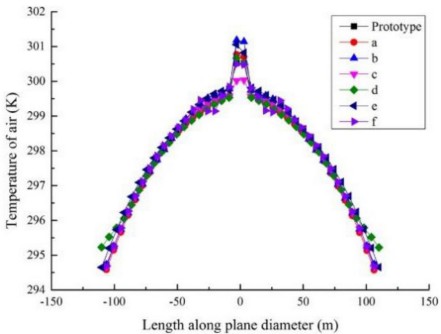

**Figure 9.** Air temperature distribution.

#### 4.1.2. The Influence on the Velocity

The relative standard deviation of velocity is usually used to quantify the flow field uniformity [44,45]. The relative standard deviation of the velocity is compared as an intuitive method of judging the degree of uniformity. The relative standard deviation of the velocity is represented by $C_v$. When $C_v \leq 0.1$, the uniformity of the airflow is optimal; when $C_v \leq 0.15$, the uniformity of the airflow is good; when $C_v \leq 0.25$, the uniformity of the airflow is qualified. The formula for calculating the relative standard deviation of the velocity $C_v$ is as follows:

$$C_v = \frac{S_v}{\overline{v}} \times 100\% \tag{13}$$

$$S_v = \sqrt{\frac{1}{n-1}\sum_{i=1}^{n}(v_i - \overline{v})^2} \tag{14}$$

Here, $C_v$—relative standard deviation of velocity; $S_v$—relative standard deviation of velocity; $v_i$—the velocity of the *i*-th measuring point, m/s; $\overline{v}$—average velocity of all measuring points; $n$—number of velocity measuring points.

A total of 480 velocity measurement points were selected in the sweeping plane of the turbine blades. The relative standard deviations of the velocities are shown in Figure 10. From Figure 10, the flow field uniformity of the SCPPS with or without baffles can be seen.

When the system without baffles, the relative standard deviation of the velocity at the turbine blades sweeping plane is 0.067, is at the state of optimal uniformity. The relative standard deviation of the velocity in the system with b-type baffles is only 0.058, and the flow field uniformity is improved by approximately 14.9%, and the air flow stability is enhanced. The relative standard deviations of the velocities in the system with the baffles of a, d, and f-type are relatively higher, 0.091, 0.098, and 0.0890 respectively. The flow field uniformity in the system with the baffles of a, d, and f-type is worse than that in the system with the other types, but is still lower than 0.1 at the optimal flow field uniformity criterion. Therefore, the baffles can intensify the flow field uniformity. The b-type baffle to SCPPS is the best in flow field uniformity.

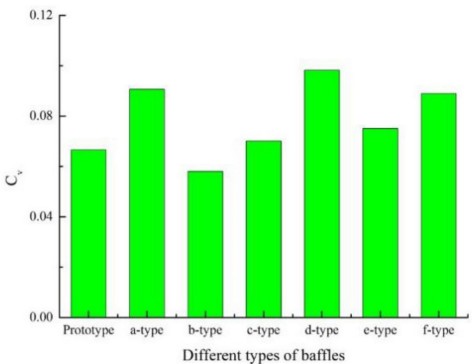

**Figure 10.** Velocity standard deviation at the turbine surface with different types of baffles.

4.1.3. The Influence on the Pressure

The pressure change has an impact on the suction effect of the chimney, which in turn affects the output power of the system. The systems with different baffles have different pressure variations at the turbine blades sweeping plane. The pressure distributions at the turbine blades sweeping plane in the SCPPS with or without different baffles are shown in Figure 11. The pressure distribution is symmetrical, but the pressure change trends are different with different types of baffles. The pressures at the turbine blades sweeping plane both in the non-baffle and the a-type baffle system continuously increase from the inner wall of the chimney to the center of the chimney. The minimum pressure is −89.889 Pa and −95.138 Pa, and the maximum pressure is −47.746 Pa and −50.996 Pa respectively. The difference is 46.9% and 46.4% respectively, and the pressure variation trend also remains similar. In SCPPS with b or c-type baffles the pressures increase from the inner wall of the chimney toward the center of the chimney, and the negative pressure near the wall is larger, respectively −78.241 Pa and −74.420 Pa, and the minimum negative pressure value in the central region is −48.792 Pa and −51.179 Pa respectively. The differences between two pressures in SCPPS with b-type or c-type are 37.6% and 31.2% respectively. In SCPPS with d, e, and f-type baffles the pressures increase first and then decrease along the inner wall of the chimney toward the center of the chimney. The pressure in the center of the chimney is −85.080 Pa, −81.219 Pa, and −74.913 Pa, respectively. The pressure near the wall is −76.824 Pa, −75.951 Pa, −71.095 Pa respectively. In addition, the pressure difference is 9.7%, 6.5% and 5.1% respectively, all less than 10%. Compared with other types, the pressure variation in these three types is the least, and there is a larger negative pressure area at the central region. In a word, the pressure distribution at the turbine blades sweeping plane is different under the influence of different types of baffles.

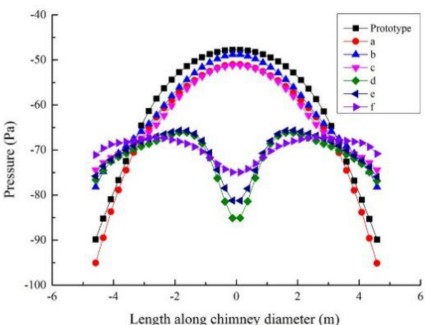

**Figure 11.** Pressure distribution of different types of baffles at the turbine blades sweeping plane.

4.1.4. The Influence on the Output Power

The influence of different types of baffles on the output power of SCPPS was calculated using Equations (11) and (12) for theoretical power and actual power. The calculation results are shown in Table 5. It can be seen from Table 5 that different types of baffles have different effects on the output power, and have different degrees of improvement compared to the output power of the system without baffles. Among the different types of baffles, the b-type baffles increase the system output power the most. The actual output power is 30.368 kW, which is 19.1% higher than the actual output power of the system without baffles. The increase of the output power of the system with c-type baffles is the least, and its actual output power is 26.602 kW, and the rate of increase is only 4.3%; and the rate of increase in the output power of the systems with the a, d, e, and f-type baffles is 14.5%, 16.6%, 15.6%, 11.2% respectively, all above 10%. It can be seen from the comparison of output power of the systems different types of baffles that the baffles have a positive effect on the increase of the system output power, and the b-type baffle has the best effect.

**Table 5.** The output powers of the SCPPS without or with different baffles.

| Baffle Type | Theoretical Power (kW) | Real Power (kW) | Percentage Increase (%) |
|---|---|---|---|
| prototype | 38.223 | 25.495 | 0 |
| a-type | 43.782 | 29.203 | 14.5 |
| b-type | 45.529 | 30.368 | 19.1 |
| c-type | 39.883 | 26.602 | 4.3 |
| d-type | 44.573 | 29.73 | 16.6 |
| e-type | 44.168 | 29.46 | 15.6 |
| f-type | 42.5 | 28.348 | 11.2 |

*4.2. The Influence of Different Numbers of b-Type Baffles*

4.2.1. The Influence on the Temperature

To compare the temperature distribution of different numbers of b-type baffles in SCPPS, the temperature contours of the ZOX plane at 1 m high from the ground is shown in Figure 12. From Figure 12, the temperature distribution in the collector is even and symmetrical. The temperature gradually rises from the entrance of the collector to the entrance of the chimney. The entrance temperature in SCPPS is about 294.7 K, and the highest temperature of the plane in SCPPS without baffles is the lowest, which is 300.5 K, while the highest temperature in SCPPS with b-type baffles is the highest, at 301.2 K, with a difference of only 0.7 K. The temperature variation is very small. It can be seen that different numbers of baffles have little effect on the temperature distribution in the collector.

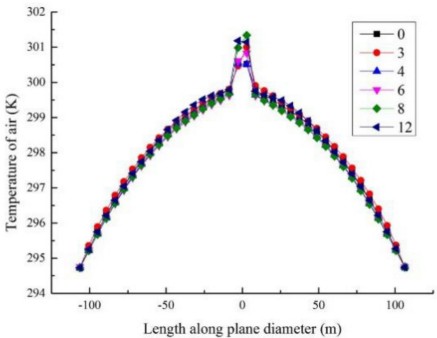

**Figure 12.** Air temperature distribution in the plane Y = 1 m.

4.2.2. The Influence on the Velocity

The relative velocity standard deviation is used to quantify the stability of the flow field, and the stability of the flow fields in the systems with different numbers of b-type baffles is compared. The results are shown in Figure 13. It is shown that as the number of baffles increases, the flow field uniformity first decreases and then increases, and the $C_v$ first increases and then decreases. When the number of baffles is 3, the relative velocity standard deviation is the largest, which is 0.0943. According to the above-mentioned $C_v$ judgment standard, the stability of the flow field in SCPPS with three baffles is at the optimal stage, but its uniformity is the worst, compared with no baffle and 12 baffles, the difference rate of both is 29.4% and 38.4% respectively. Therefore, it is observed that the number of baffles has an expressively influence on the stability of the flow field, and the $C_v$ with 12 baffles is optimal.

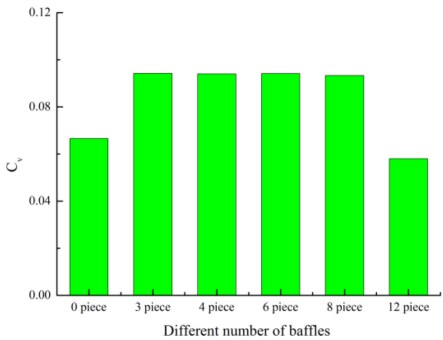

**Figure 13.** Velocity standard deviation at the turbine surface with different types of baffles.

4.2.3. The Influence on the Pressure

The pressure distribution at the turbine blades sweeping plane in SCPPS with different numbers of b-type baffles is shown in Figure 14. It can be clearly seen from the figure that the pressure variation is almost the same, and the pressure increases from the inner wall of the chimney toward the center of the chimney. Compared with the pressure distribution in SCPPS without a baffle, the maximum negative pressure at the turbine blades sweeping plane in SCPPS with baffles is lower than that of prototype. When the 12 pieces of baffles are used, the maximum value of negative pressure is −78.241 Pa, and the maximum value of negative pressure of prototype is −89.837 Pa, with a difference of about 12.9%. Nevertheless, different numbers of baffles have little effect on the pressure distribution at the turbine blades sweeping plane. The maximum negative pressure does not exceed 5%, and the minimum negative pressure does not exceed 1%. The overall pressure distributions are very similar.

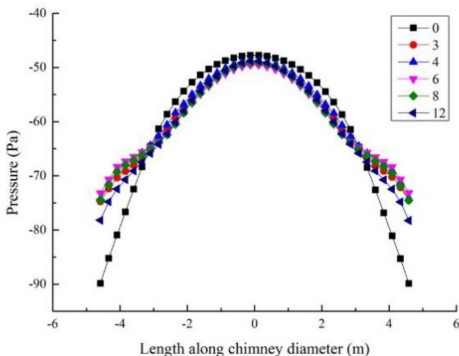

**Figure 14.** The pressure distribution at turbine plane in SCPPS with different numbers of b-type baffles.

#### 4.2.4. The Influence on the Output Power

As mentioned earlier, it can be seen that in SCPPS with b-type baffles has the best effect on the output power. Equations (11) and (12) are used to calculate the theoretical output power and actual output power of SCPPS with different number b-type baffles. The calculation results are shown in Table 6. Compared with prototype, it can be found that as the number of baffles decreases, the degree of power increase gradually weakens. When the number of baffles is 4, the system output power is only increased by 2.3%, which is the lowest. When the number of baffles is 12, the output power is increased by 19.1%, which is the highest. When the numbers are 8, 6, and 3, the increase in the output power is 8.1%, 4.9%, and 4.6% respectively, and the increment is less than 10%. The output power of the SCPPS can be intensified with different baffles. With the b-type baffles, when the number is 12, the most output power can be obtained. We can see that the more baffles in SCPPS, the most output power generated.

**Table 6.** The output power of SCPPS with different number of b-type baffles.

| Baffle Number (Pieces) | Theoretical Power (kW) | Real Power (kW) | Percentage Increase (%) |
|---|---|---|---|
| prototype | 38.223 | 25.495 | 0 |
| 12 | 45.529 | 30.368 | 19.1 |
| 8 | 41.305 | 27.55 | 8.1 |
| 6 | 40.103 | 26.749 | 4.9 |
| 4 | 39.09 | 26.073 | 2.3 |
| 3 | 39.989 | 26.673 | 4.6 |

## 5. Analysis of Experimental Results

### 5.1. Experimental Conditions

To lower the influence of outdoor environmental factors on the experiment results, the experimental research was conducted under the steady conditions of radiations, ambient temperatures, and outdoor wind velocities. The data from environmental parameters such as average ambient temperature, average wind velocity and average solar radiation intensity measured in the experiments are shown in Figure 15.

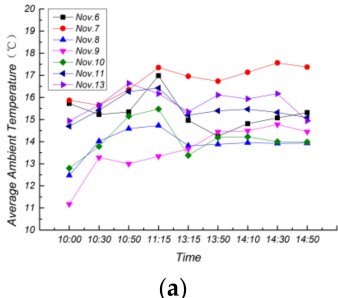 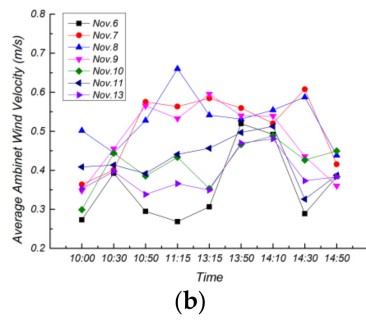 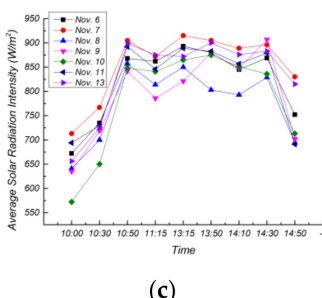

(**a**)          (**b**)          (**c**)

**Figure 15.** The ambient experimental data. (**a**) Average ambient temperature; (**b**) Average ambient wind velocity; (**c**) Average solar radiation intensity.

It can be seen from Figure 15 that in different experimental days, the data of the average ambient temperatures, average ambient wind velocities, and average solar radiation intensities in the four time periods are relatively similar. The maximum differences of the average ambient temperature or average wind velocities do not exceed 20%. The maximum difference on average solar radiation intensities is 16.7%, and the minimum difference is 10.8%. The free outdoor experimental environment is regarded as the same environment, and the experimental data in some period is taken for analysis. The taken experimental stages were classified into A, B, C and D-period. The experimental periods are shown Table 7. The pressure differences we tested are small, so we will not discuss the pressure here.

**Table 7.** Experimental Periods.

| Experiment Period Number | A-Period | B-Period | C-Period | D-Period |
|---|---|---|---|---|
| Experiment period | 10:30–10:46 | 13:50–14:06 | 14:10–14:26 | 14:50–15:06 |

*5.2. Analysis of Different Types of Baffles*

The velocity distribution at the bottom of the chimney is shown in Figure 16. As it can be seen from the figure, the measured velocity distributions in SCPPS with the different types of baffles are very different. The velocity of b-type baffles is concentrated in the range of 1.2 m/s above. In the experiment conducted in period B, the velocity distribution in SCPPS with b-type baffles in the range above 1.2 m/s accounted for 64.6%, and the velocity distribution in SCPPS with prototype, a-type and c-type baffles is 42.4%, 17.2%, and 38.4% respectively, which are much lower than that in SCPPS with the b-type baffle. In the A, C and D periods, the velocities in SCPPS with b-type baffles above 1.2 m/s accounts for 40.4%, 52.5%, 49.5%, respectively, which are higher than the velocities in other cases.

The temperature distribution were measured by eight measurement points in the experimental apparatus. The temperature distributions in SCPPS with different types of baffles are shown in Figure 17. It can be seen from the figure that the temperatures gradually increase from the entrance of the collector to the bottom of the chimney. The highest average temperature appears at the bottom of the chimney. The b-type baffle can make the temperature at the bottom of the chimney increase most significantly, and the highest average temperature appears in period B, which can reach 28.8 °C. Daily solar radiation, ambient temperature and wind velocity change constantly, the temperatures vary continuously in the system. However, all the temperature distribution trends remain the same, which is verified the simulation. Obviously, the system with the b-type baffles has more prominent heat transfer characteristics in the experiment. The improvement effect of b-type baffles for SCPPS on air heat transfer characteristics is more obvious than others in the experiment.

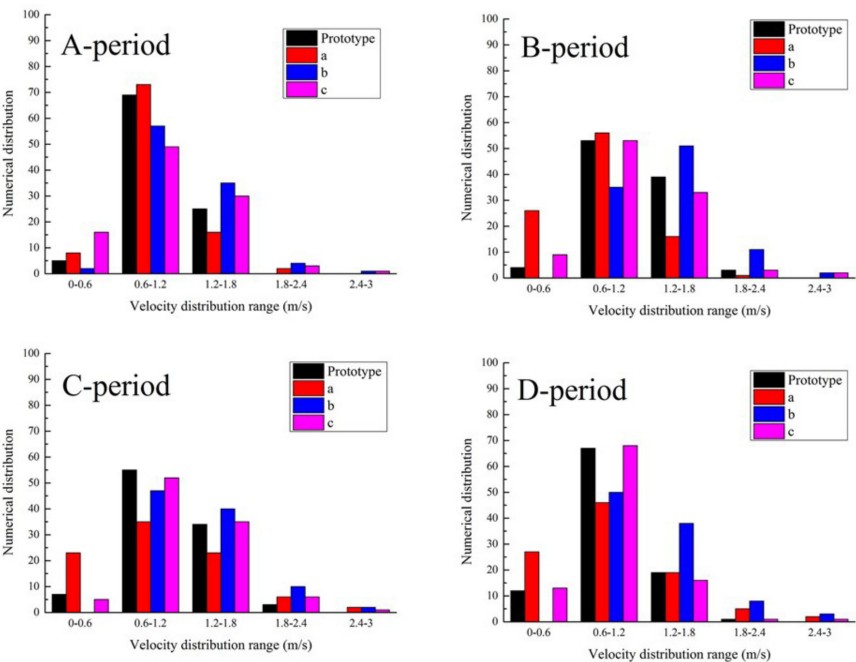

**Figure 16.** The velocity distribution at the bottom of the chimney in SCPPS with different types of baffles at different periods.

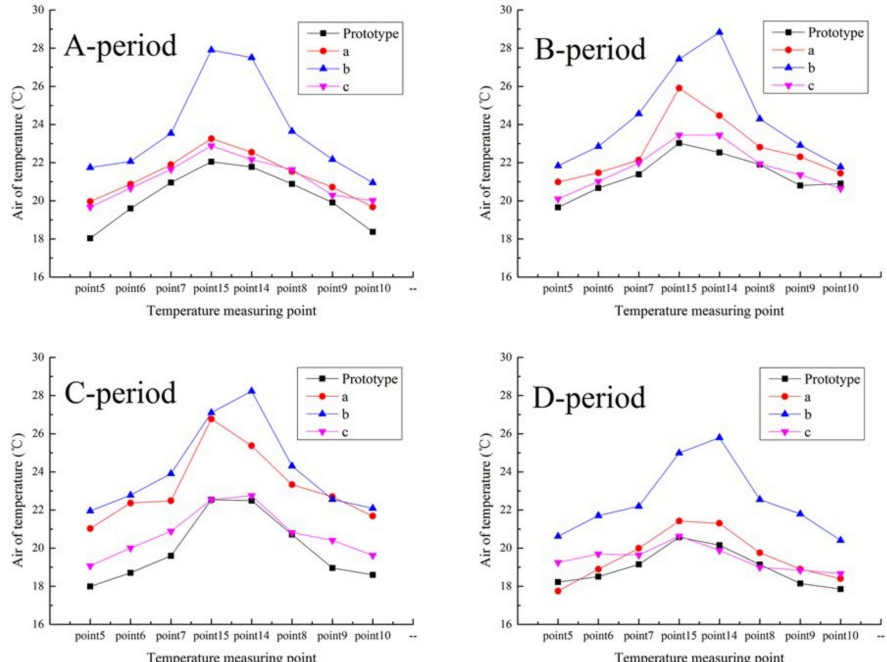

**Figure 17.** Temperature distributions of different types of baffles in the collector at different time periods.

### 5.3. Analysis of Different Numbers of Baffles

According to experimental data, different numbers of b-type baffles have different effects on the velocity changes at the bottom of the chimney, and the velocity distribution is shown in Figure 18. It can be concluded that in period B and C, the velocities in the system with 12 b-type baffles are concentrated above 1.2 m/s, accounting for 64.6% and 52.5%, and when the number of baffles is 0, 3, 6, and 8, accounting for 42.4% and 37.4%, 35.4% and 29.3%, 53.5% and 47.5%, and 8.1% and 47.5%, respectively. In period A, the velocities in

the system with 3 b-type baffles are concentrated above 1.2 m/s accounting for 55.6%. At this time, when the number of b-type baffles is 0, 6, 8, and 12, the velocities are 1.2 m/s or more accounted for 25.3%, 50.5%, 23.2%, and 40.4%. In period D, the velocity distribution in the system with 6 b-type baffles is above 1.2 m/s accounting for 57.6%. Meanwhile, when the number in the system with b-type baffles is 0, 6, 8, and 12, the velocities are 1.2 m/s or more accounted for 20.2%, 15.2%, 40.4%, and 49.5%. As mentioned above, it can be seen that the different experimental time and numbers of baffles have different effects on the velocity at the bottom of the chimney. Under the condition of weak solar radiation, 3 or 6 b-type baffles can increase the concentrated distribution of the velocities at the bottom of the chimney, which means increasing the velocities. In the case of strong solar radiation, 12 b-type baffles are more conducive to the velocity increment at the bottom of the chimney. Compared with other numbers in the system with b-type baffles, 12 b-type baffles can force the system maintain a relatively uniform and stable velocity distribution all the experimental time.

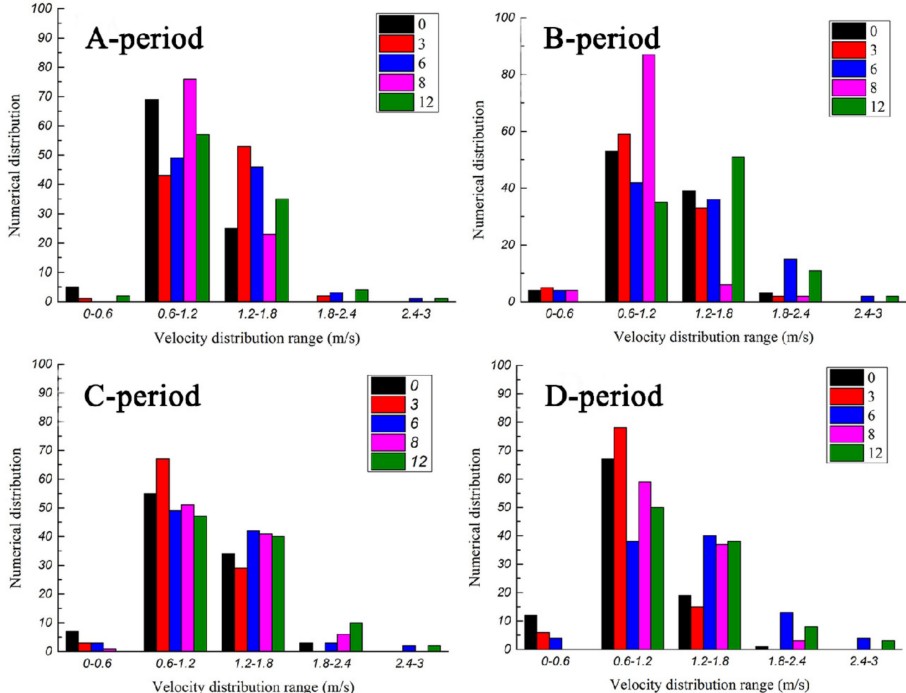

**Figure 18.** The velocity distribution at the bottom of the chimney in SCPPS with different numbers of baffles in different time periods.

The temperatures measured in SCPPS are shown in Figure 19. It can be analyzed that the temperature distribution in the system with 12 b-type baffles is higher than that in the system with other baffle numbers. The temperature distribution gradually rises from the entrance of the collector to the bottom of the chimney. During period C, the average temperature at the bottom of the chimney reached the highest, which was 28.8 °C.

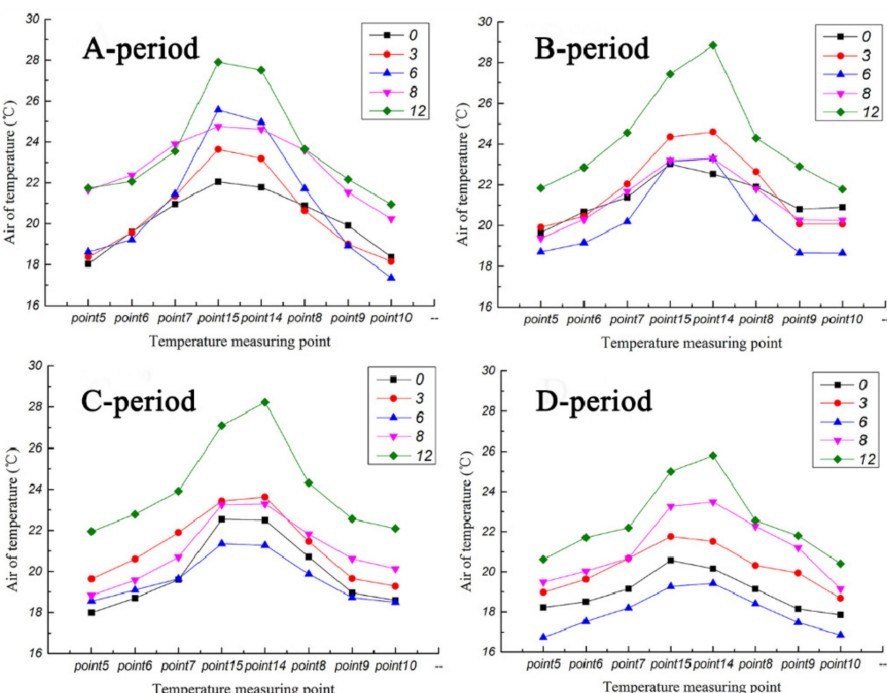

**Figure 19.** Temperature distribution in the system with different numbers of baffles in different time periods.

## 6. Conclusions

Six different types of baffles were designed and corresponding different SCPPS models were established. The numerical simulations were carried out by combining with the Solar Tracking algorithm to study the heat transfer and flow characteristics of the SCPPS with different baffles. Combined with the experimental verification, the best type and number of baffles that improve the performance of the system are identified.

(1) The research is aimed the SCPPS with non-baffles(prototype) and six different types of baffles respectively on the influence of the pressure field, temperature field, velocity field, the relative standard deviation of velocity and output power of SCPPS. It is concluded that the addition of the six different baffles in SCPPS can increase the velocity, temperature, pressure field to varying degrees, but the b-type baffles (the baffles are inner, half-size, straight) can improve the temperatures and velocity uniformity of the system, and the output power is promoted significantly.

(2) For b-type baffles, five different baffle numbers (3, 4, 6, 8, and 12 pieces, respectively) of SCPPS models were carried out, and it was obtained that among the five different baffle numbers, the improvements of the temperature, pressure, velocity uniformity and output power of the SCPPS with 12 baffles are optimal. It can be seen that the greater the number of baffles, the more obvious the improvement in system performance, but the higher the cost of the corresponding materials.

(3) Through experiments, it is shown that the velocity distribution in the chimney with b-type baffles is higher than that in the chimney other types of baffles. The b-type baffles increase the temperature at the bottom of the chimney most significantly. In the case of weak solar radiation, 3 or 6 b-type baffles can increase the concentrated distribution of the velocities at the bottom of the chimney. With strong solar radiation, 12 b-type baffles are more conducive to the concentrated distribution of the velocities at the bottom of the chimney. Compared with other numbers of b-type baffles, a relatively uniform and stable velocity distribution can be shown in SUPPS with 12 b-type baffles in the whole day. The temperature distribution of the system with 12 b-type baffles is higher than that of the system with other numbers of baffles. The

experimental conclusions are basically in keep with the simulation results, which verifies the simulation.

**Author Contributions:** H.W. conceived and designed the experiments; J.C. and F.Z. performed the experiments; J.C. and P.D. analyzed the data; Q.L. supervision for this work; H.W. and J.C. wrote the paper. All authors have read and agreed to the published version of the manuscript.

**Funding:** This work is supported by the Key Research and Development Program of Shandong Province, China (Grant number 2019GGX104034).

**Institutional Review Board Statement:** Not applicable.

**Informed Consent Statement:** Not applicable.

**Data Availability Statement:** Not applicable.

**Acknowledgments:** Funding from the Key Research and Development Program of Shandong Province is gratefully acknowledged; the authors wish to acknowledge my friends Tao Li, Haisheng Bi, Hailong Chen, Xing Cao, Fei Wang, Qingdao University of Science&Technology, for their helps in the course of writing this paper.

**Conflicts of Interest:** The authors declare no conflict of interest.

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
