# Peer review of "Simulation and Experimental Study of the Influence of the Baffles on Solar Chimney Power Plant System"

_processes, doi:10.3390/pr9050902_

Round 1

Reviewer 1 Report

1. It is said that the numerical analysis was performed using ANSYS fluent, but
please write whether the 2d condition was used or the 3d condition was used. In
addition, it is necessary to check whether the governing equation is set according
to 2D and 3D.
2. According to the additional explanation condition (1) for Equation (4) in page 4
/ Mathmatical model, 
   when it is in a steady state, it is necessary to modify
Equation (1) or additional explanation.
3. (Page 4 - Mathmatical model) please write in detail what the exact meaning of
condition (4) is. [The material properties of the material]
4. (Page 4) Continuity, Momentum, Energy, Turbulent Kinetic equation, please write
a reference.
5. (Page 5 - 2.3 Boundary Condition) there are many turbulence models in 'ANSYS
fluent'. Please add the reason for selecting the    standard wall function and the
comparison of the results when calculated with other turbulence models.
6. Added reference data on (page 10 - The relative standard deviation of velocity).
7. The size of the picture is too small to see, so it would be nice to change it
large.
8. Classifying superscripts when indicating units ex) 8page / W/m2 -> W/m2
9. Make the SI prefix notation clear ex) KW -> kW
10. When displaying the comparison group on the graph, mark pieces or type to
make it easier for readers to understand.
11. Make sure to unify uppercase and lowercase comparison groups when
indicating type. [ex) a type, A type]

Author Response

We would like to express our sincere thanks to the reviewers for the constructive and positive comments.

Reviewer 2 Report

The English of the article needs proofreading, because in its current form, serious mistaked make the text often unclear, like:
"can be changed into generate", "heat collector", "heat resource", etc.
The literature review is to be reorganised, because it should not be categorised by researchers, but by different aspects of technical content.
Number of references (21) is too low and they must report on results from all over the globe.
In Figure 1 the dimensions are very badly visible and the figure quality is poor.
Assumption (4) about material properties is unfinished.
A nomenclature section is to be used instead of explaining each equation one by one. Abbreviations like SC remain unexplained in some cases too.
In the nomenclature section in the end Rayleigh and Reynolds number are confused.
It is not possible to measure or to simulate 879 or 877 kW/m2 solar radiation intensity.
kW is small k and capital W.
It remains unclear why the dots in Figures 9 and 12 are connected.

Author Response

(The authors gave the same response as above.)

Reviewer 3 Report

SUMMARY: The work investigates different types and numbers of baffles needed to improve the power generation efficiency of the SUPPS (Solar Updraft Power Plant System). The effect of different types and numbers of baffles on the velocity, pressure, temperature, and output power are determined. Findings claim that the results from the model-based simulation and those from experiments are agreeable.
OVERALL COMMENTS: The problem chosen seems relevant. However, the formulation of the problem and methodology need significant revision. In particular, the following need to be explicated clearly: What exactly is the problem statement, its objective, and the decision variables involved? Further, the authors need to proofread the whole manuscript and fix all inconsistencies. Detailed comments are provided below:
CONTRIBUTIONS:
• What exactly are the novel contributions of the work in relation to the state-of-the-art? Novelty needs to be highlighted – preferably a paragraph explicating the contributions.
GENERAL:
• The manuscript needs to be proofread to avoid shoddy writing. As examples, consider the below:
• L119: “To simple the calculation...” – likely simplify instead of simple.
•  L120: “The system is steady state” – do you mean the system is at steady state?
•  L120-124: Three of the four items in the list don’t end with ‘.’ unlike (3) – lack of consistency.
•  L458-466: “The research … outstanding” – this sentence is either missing full stops ‘.’ or contains 120 words.
•  L467-469: “… and it was obtained that 12 inner half-size straight baffles were aligned.” – what is this supposed to mean?
TECHNICAL: The following technical concerns need to be addressed:
•  L125-127: Equations 1, 2, 3, 6, and 7 are stated without explanation and the authors do not even state what the symbols mean and how they relate to this work. Are the readers expected to understand the work?
•  L153-154: “The coupling of pressure and velocity is calculated by SIMPLE algorithm” – is SIMPLE an acronym? The authors can provide more context.
• What exactly is being modeled – what are the model inputs and outputs? A single paragraph to summarize exactly what the model accepts as inputs and is meant to output. Alternatively, the authors can consider a diagrammatic representation to display model inputs, parameters, and outputs.
•  Is there a reason to not consider significance analysis to be able to claim statistical significance of the simulation/experimental data? Without determining statistical significance, it would be
Overall, the problem formulation, and methodology seem unclear and difficult to follow, and the manuscript needs a significant improvement focusing on clarity of presentation and writing style.

Author Response

(The authors gave the same response as above.)

Reviewer 4 Report

The authors have investigated  a promising  solar updraft power plant system with bank of baffles  arranged under the heat collector  and providing a mathematical  formulation and the simulation of the systems with 5 different baffle numbers with the experimental apparatus.
The basic sections (Introduction, Conclusion, Literature Cited, etc.)
are adequate and without English correction.
However I suggest strictly to include in the references section the
publication already submitted:

Sciuto, G. L., Cammarata, G., Capizzi, G., Coco, S., & Petrone, G. (2016, June). Design optimization of solar chimney power plant by finite elements based numerical model and cascade neural networks. In 2016 International Symposium on Power Electronics, Electrical Drives, Automation and Motion (SPEEDAM) (pp. 1016-1022). IEEE.

 The present work may be accepted with minor revision.

Author Response

(The authors gave the same response as above.)
